# Better to Teach than to Give: Domain Generalized Semantic Segmentation via Agent Queries with Diffusion Model Guidance

**Fan Li** [1]  **Xuan Wang** [1]  **Min Qi** [1]  **Zhaoxiang Zhang** [1]  **Yuelei Xu** [1]

## Abstract

Domain Generalized Semantic Segmentation (DGSS) trains a model on a labeled source domain to generalize to unseen target domains with consistent contextual distribution and varying visual appearance. Most existing methods rely on domain randomization or data generation but struggle to capture the underlying scene distribution, resulting in the loss of useful semantic information. Inspired by the diffusion model's capability to generate diverse variations within a given scene context, we consider harnessing its rich prior knowledge of scene distribution to tackle the challenging DGSS task. In this paper, we propose a novel agent **Query**-driven learning framework based on **Diff**usion model guidance for DGSS, named QueryDiff. Our recipe comprises three key ingredients: (1) generating agent queries from segmentation features to aggregate semantic information about instances within the scene; (2) learning the inherent semantic distribution of the scene through agent queries guided by diffusion features; (3) refining segmentation features using optimized agent queries for robust mask predictions. Extensive experiments across various settings demonstrate that our method significantly outperforms previous state-of-the-art methods. Notably, it enhances the model's ability to generalize effectively to extreme domains, such as cubist art styles. Code is available at https://github.com/FanLiHub/QueryDiff.

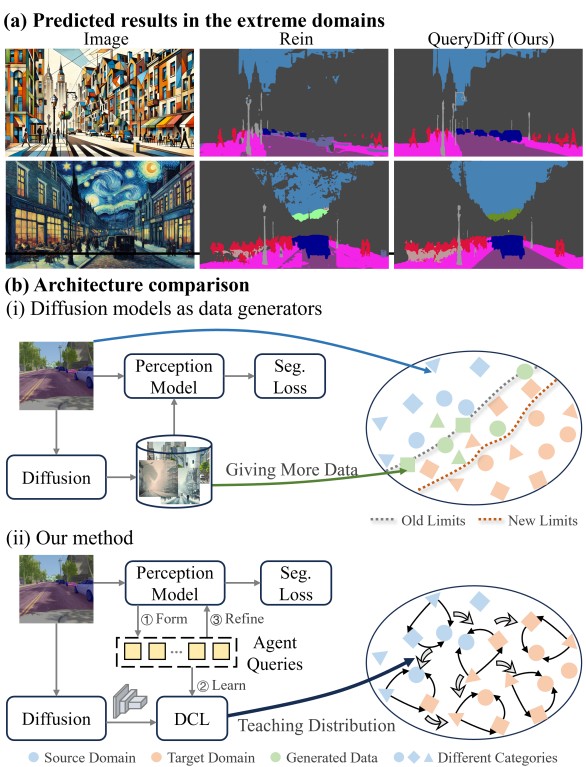

*Figure 1.* (a) The segmentation results are predicted by our method and previous SOTA approach in extreme domains, including cubist art and post-impressionist styles generated by ChatGPT. (b) Previous methods use the diffusion model as a data generator, which struggles to cover all variations in the target domain, resulting in limited performance. Our method employs agent queries to learn scene distribution knowledge from the diffusion model, capitalizing on the inherent consistency of this distribution across domains to improve segmentation model generalization.

## 1. Introduction

Semantic segmentation, a fundamental task in computer vision, has seen significant progress in recent years (Long et al., 2015; Chen et al., 2017; Fu et al., 2019; Zhong et al.,

[1]Northwestern Polytechnical University, China. Correspondence to: Yuelei Xu <xuyuelei@nwpu.edu.cn>.

*Proceedings of the 42nd International Conference on Machine Learning*, Vancouver, Canada. PMLR 267, 2025. Copyright 2025 by the author(s).

2020; Cheng et al., 2022). However, it experiences considerable performance degradation when there is a gap between the training data and the test data. This has sparked interest in Domain Generalized Semantic Segmentation (DGSS) (Zhao et al., 2022; Jia et al., 2023; Li et al., 2023). DGSS aims to train a model to learn generalizable features from the source domain, enabling it to perform well on a variety of unseen target domains with consistent contextual distribution but significantly varying visual appearances.

The majority of DGSS methods are built upon Domain Randomization (DR) (Wang et al., 2020; Zhao et al., 2022; Zhong et al., 2022; Jiang et al., 2023), which diversifies source domain images through photometric and geometric transformations. Although somewhat effective, such transformations are constrained by the structure and content of the source data, failing to create new contextual dependencies between instances within the scene. This results in limited diversity that fails to encompass broader scene distributions, i.e., the spatial arrangement and contextual dependencies implicit in the inter-instance relationships (e.g., cars on roads, buildings aligned along streets) that enable the model to move beyond isolated instances and grasp unified objective laws across different scenario domains. As a result, the model remains robust only to specific patterns of variation, resulting in poor generalization performance.

Recently, diffusion models have opened up new avenues for DGSS with remarkable capabilities in capturing complex scene distributions to generate high-quality, realistic samples (Dhariwal & Nichol, 2021; Ho et al., 2020; Rombach et al., 2022). Building on this, several studies have utilized the scene distribution priors of diffusion models to expand the semantic distribution of source domain data. For instance, DIDEX (Niemeijer et al., 2024) employs a diffusion model to generate a pseudo-target domain for domain extension. DatasetDM (Wu et al., 2023a) presents a generic dataset generation model capable of producing diverse images and corresponding annotations using diffusion models. DGInStyle (Jia et al., 2025) crafts diverse task-specific images by sampling the rich prior of a pretrained diffusion model while maintaining strict adherence to semantic layout conditions. While these methods have achieved notable improvements in model performance, expanded data remains inadequate to support the almost infinite distributional variations in target scenarios, as shown in Figure 1(b). Moreover, expanding the training data substantially incurs significant computational costs and time consumption for the model. Adhering to the principle that *teaching* a man to fish is better than *giving* him a fish, a natural curiosity has been raised: **Can the powerful scene distribution priors embedded in diffusion models be directly leveraged to enhance segmentation generalization?**

A straightforward solution is to harness the diffusion model as a general knowledge extractor to enhance DGSS performance by extracting diffusion features and mapping them to segmentation results. However, this comes with two new challenges: 1) *The iterative sample process in diffusion models involves multiple steps of denoising, which is computationally expensive and time-consuming, making it inefficient for perception tasks.* 2) *Diffusion features encompass not only high-level semantic information but also low-level visual appearance details (e.g., color and texture), which, though crucial for generation tasks, are unnecessary*

*and even detrimental for perception tasks*.

To address all these challenges, in this work, we propose QueryDiff, an agent queries-driven framework guided by diffusion models for DGSS, which utilizes the powerful scene distribution priors embedded in diffusion models to enhance semantic segmentation generalization. Specifically, for **the first challenge**, QueryDiff employs learnable queries to interact with multi-scale image features from the segmentation backbone, aggregating instance-level semantic information to form agent queries that serve as the interface throughout the pipeline, instead of directly relying on diffusion features. Next, guided by the diffusion features, the agent queries learn to construct the semantic distribution of the scene. To address **the second challenge**, we propose diffusion consistency loss (DCL) that aims to eliminate intricate visual appearance details from the diffusion features, allowing the agent queries to focus on comprehending scene distribution for generalized semantic representations. Ultimately, we use these agent queries to refine the features in the segmentation decoder, integrating the prior knowledge of scene distribution from the diffusion features into the segmentation model, and thereby producing more robust predictions. Extensive experiments across various DGSS settings demonstrate that our method surpasses existing approaches, achieving state-of-the-art performance. Moreover, as shown in Figure 1(a), our method generalizes effectively to extreme domains, such as cubist and impressionist paintings. The contributions are summarized as follows:

- We propose a novel agent queries-driven learning framework named QueryDiff, which employs agent queries as the interface to mine the scene distributions embedded in diffusion features, thereby improving the robustness of the model with respect to domain shifts.

- We propose diffusion consistency loss (DCL) to avoid intricate visual details in diffusion features from interfering with agent queries, enabling them to better focus on learning the semantic distribution of the scene.

- QueryDiff is a concise framework with high performance. It outperforms a wide variety of baselines and reaches a new state-of-the-art performance on extensive DGSS benchmarks.

## 2. Related Work

### 2.1. Diffusion Model

Diffusion models (Ho et al., 2020; Dhariwal & Nichol, 2021; Rombach et al., 2022) have recently demonstrated state-of-the-art image generation quality. With the success of diffusion models in generative tasks, some pioneer works have explored the application of diffusion models to various visual perception tasks (Xu et al., 2023; Tang et al.,

2023; Ji et al., 2023; Lee et al., 2024; Ke et al., 2024). For example, Wu et al. (2023b) exploit cross-attention of the diffusion model to localize class-specific regions and generate a high-resolution segmentation mask. Xu et al. (2023) introduce ODISE, which integrates the internal representations of pre-trained diffusion models and discriminative models to effectively perform panoptic segmentation across any category in the wild. Ji et al. (2023) propose DDP, an extension of denoising diffusion process for semantic segmentation and depth estimation, transforming noise samples into desired predictions iteratively guided by input images. These methods rely on the diffusion model as the backbone, but its iterative sampling process, involving multiple denoising steps, incurs significant computational overhead and time consumption, rendering it inefficient for perception tasks. Thus, our method leverages agent queries as an interface to mine the underlying scene distribution embedded in diffusion features and refine the features of the segmentation model to improve its robustness, avoiding the direct involvement of diffusion features in the segmentation process.

### 2.2. Domain Generalized Semantic Segmentation

Domain generalized semantic segmentation (DGSS) aims to enhance model robustness against domain shifts. The current approaches fall into two categories: normalization/whitening and domain randomization. Normalization/whitening methods learn domain invariant features by tailor-made modules to remove domain-specific features (Pan et al., 2018; Choi et al., 2021; Peng et al., 2022; Xu et al., 2022), but they struggle with complex domain shifts. Domain randomization seeks to broaden the distribution of source domain by either diversifying the data style through data augmentation (Lee et al., 2022; Zhao et al., 2022; Kim et al., 2023) or enriching the data content through generating new data (Benigmim et al., 2024), offering improved generalization performance. Recent studies have investigated the use of visual foundation models in DGSS. Benigmim et al. (2024) integrate various foundation models, including employing the diffusion model as a data generator, leveraging the robust features of CLIP, and utilizing SAM to refine pseudo labels for self-training. Jia et al. (2023) propose DGInStyle, a controlled pipeline for generating task-specific images with widely varying appearance data that samples the rich prior of a pre-trained diffusion model. Niemeijer et al. (2024) introduce a novel domain extension method utilizing a diffusion model to generate a pseudo-target domain with diverse text prompts and then training a generalizing model by adapting to this pseudo-target domain. Despite these advancements, current approaches leverage diffusion primarily as an auxiliary tool of data generation within training pipelines. In contrast, this paper introduces an agent queries-driven approach to fully mine domain-invariant semantic information of the diffusion features, thereby enhancing the generalization capability of semantic segmentation.

### 2.3. Learnable Query Design

Recently, a series of learnable query-based frameworks inspired by DETR (Carion et al., 2020) have been proposed. For instance, Panoptic SegFormer (Li et al., 2022) introduces a query decoupling strategy to prevent mutual interference between thing queries and stuff queries. MaskFormer (Cheng et al., 2021) and Mask2Former (Cheng et al., 2022) solve both semantic- and instance-level segmentation tasks in a unified framework by employing object queries to aggregate pixels within the same semantic region. MP-Former (Zhang et al., 2023) accelerates training by feeding noisy ground truth masks and learnable queries into the Transformer decoder to reconstruct the originals. Rein (Wei et al., 2024) fine-tunes vision foundation models by incorporating a set of randomly initialized queries as learnable parameters. While these studies have shown that specific learnable queries improve performance and functionality in perception tasks, research on learnable queries for DGSS remains unexplored. Our work aims to address DGSS by designing agent queries, constructed through learnable queries, to develop a concise, general, and efficient framework to improve the generalization of the model.

## 3. Preliminary

**Problem Definition.** The goal of Domain Generalized Semantic Segmentation (DGSS) is to train a segmentation model $\varphi = g \circ \upsilon$, where $g$ and $\upsilon$ denote the backbone and decoder, respectively, on a labeled source domain $\mathcal{S}$, enabling it to generalize effectively to an unseen target domain $\mathcal{T}$. Both domains share the same set of $K$ categories.

**Stable Diffusion.** Stable diffusion involves a forward diffusion process that adds noise to the data and a reverse process that converts the noisy samples into raw data. The forward diffusion process is defined as:

$$q\left(z_t \mid z_0\right) := \mathcal{N}\left(z_t \mid \sqrt{\bar{\alpha}_t} z_0, \left(1 - \bar{\alpha}_t\right) I\right) \qquad (1)$$

where $z_0 = \mathcal{E}(x)$ and $\mathcal{E}$ is a pre-trained encoder that encodes image $x \in \mathbb{R}^{H \times W \times 3}$ into a latent representation space. $\bar{\alpha}_t := \prod_{s=0}^{t} \alpha_s = \prod_{s=0}^{t}\left(1 - \beta_s\right)$ and $\beta_s$ represents the pre-defined noise schedule (Ho et al., 2020). This process transforms data sample $z_0$ to a latent noisy sample $z_t$, and $t$ is the diffusion step. Generally, a larger $t$ corresponds to larger noise weights. The reverse process is a denoising procedure that gradually transforms the noisy samples into raw data. A step in the reverse process can be defined as:

$$p_\theta\left(z_{t-1} \mid z_t\right) := \mathcal{N}\left(z_{t-1} \mid \mu_\theta\left(z_t, t\right), \Sigma_\theta\left(z_t, t\right)\right) \qquad (2)$$

where the mean $\mu_\theta$ is predicted by noise predictor $\epsilon_\theta(\cdot)$ and the covariance $\Sigma_\theta$ is generally fixed as a predefined value.

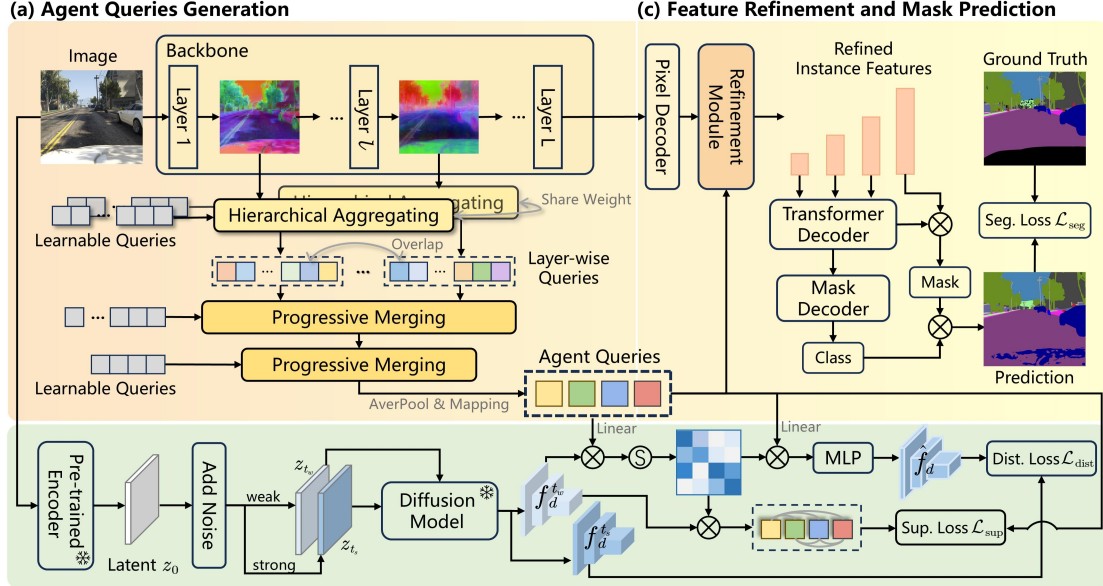

*Figure 2.* A brief illustration of our proposed framework. First, we use learnable queries to aggregate hierarchical instance features from the segmentation backbone, progressively merging them to form agent queries. Next, we utilize agent queries to learn the scene distribution information embedded in the diffusion features, optimizing their semantic representations through diffusion consistency loss (DCL) that removes visual appearance information irrelevant to the perceptual task from the diffusion features. Finally, we use the optimized agent queries to refine the instance features of the segmentation decoder and output the prediction mask.

## 4. Methodology

With their powerful ability to generate high-quality samples, diffusion models have been successfully applied to improve segmentation generalization by expanding source domain data, drawing on their rich priors of scene distribution. Guided by the principle that *teaching* a man to fish is better than *giving* him a fish, a natural question arises: Can the scene distribution priors of diffusion models be directly leveraged to enhance segmentation generalization? As illustrated in Figure 2, we propose a novel agent query-driven DGSS framework QueryDiff based on diffusion models, which consists of three key steps: (1) aggregating semantic information within a scene to generate agent queries (Section 4.1), (2) leveraging agent queries to mine scene distribution knowledge encoded in diffusion features (Section 4.2), and (3) reinforcing scene distribution information in the segmentation features via agent queries (Section 4.3).

### 4.1. Agent Queries Generation

Considering the substantial inefficiency resulting from the computationally intensive iterative sampling process of diffusion models, our approach employs agent queries as the interface throughout the framework, instead of directly incorporating diffusion features into the segmentation process. Specifically, we aggregate hierarchical visual features from the segmentation backbone and progressively merge them into a unified representation, referred to as agent queries.

**Aggregating Hierarchical Visual Features.** We establish multiple groups of learnable queries, each corresponding to a feature layer in the segmentation backbone. For the features $f_{seg}^l$ produced by the $l$-th layer of the segmentation backbone $g$, we first construct layer-wise queries as follows:

$$q_{layer}^l = \text{MLP}(\text{Softmax}(f_{seg}^l \times q_{init}^l)^{\text{T}} \times f_{seg}^l) \quad (3)$$
$$\text{with } q_{layer} = \{q_{layer}^l \in \mathbb{R}^{r \times c} \mid 1 \leq l \leq L\}$$

where $q_{layer}^l$ are layer-wise queries for $l$-th feature $f_{seg}^l$, $q_{init}^l \in \mathbb{R}^{c \times r}$ are randomly initialized learnable queries. $c$ represents the dimension of each feature layer, $r$ denotes the sequence length of $q_{init}^l$, $L$ represents the total number of feature layers in the segmentation backbone. All layers share the same MLP weights.

**Progressive Merging for Unified Representation.** Each layer-wise query corresponds to at least one instance of interest in the scene, thus there is information overlap among the layer-wise queries. To ensure that each query ultimately retrieves the features of the same instance in different layers, we merge the layer-wise queries into a whole using a progressive merging strategy. Specifically, for each stage $i$, we employ learnable queries $q_{init}^i$, map them to queries $\mathbf{Q}^i$ and the outputs of the previous stage, $q_{stage}^{i-1}$, are respectively projected into the keys $\mathbf{K}^i$ and values $\mathbf{V}^i$

$$\mathbf{Q}^i = q_{init}^i \mathbf{W}_{\mathcal{Q}}^i, \ \mathbf{K}^i = q_{stage}^{i-1} \mathbf{W}_{\mathcal{K}}^i, \ \mathbf{V}^i = q_{stage}^{i-1} \mathbf{W}_{\mathcal{V}}^i$$
$$q_{init}^i \in \mathbb{R}^{n_i r \times c}, \ q_{stage}^{i-1} \in \mathbb{R}^{n_{i-1} r \times c}, \ 1 \leq i \leq M \quad (4)$$

where $\mathbf{W}_{\mathcal{Q}}^i$, $\mathbf{W}_{\mathcal{K}}^i$ and $\mathbf{W}_{\mathcal{V}}^i$ are linear projections. $q_{init}^i$ are randomly initialized learnable queries, $n_i$ denotes the number of the queries produced at the $i$-th merge stage, $M$ represents the total number of merging stages and $q_{stage}^0 = q_{layer} \in \mathbb{R}^{Lr \times c}$. Next, we merge the smaller groups into larger ones based on the similarity matrix in the embedding space:

$$\hat{q}_{stage}^i = \text{FFN}\left(\frac{\exp\left(s^i\right)}{\sum_{j=1}^{hw} \exp\left(s^i\right)} \times \mathbf{V}^i\right), \ s^i = \frac{\mathbf{Q}^i(\mathbf{K}^i)^T}{\sqrt{d^i}} \tag{5}$$

where $\hat{q}_{stage}^i \in \mathbb{R}^{n_i r \times c}$ are the output of the current stage, $d^i$ is a scaling factor, and FFN consists of a linear mapping followed by an activation layer. $h$ and $w$ denote the height and width of the similarity matrix, respectively. Notably, the number of queries generated at each merge stage decreases progressively, i.e., $n_{i+1} < n_i$. Next, $\hat{q}_{stage}^i$ are fed to self-attention and FFN to enrich the representation of each query and obtain the output $q_{stage}^i$, which is the next stage of inputs. After the final stage, $M$, we average their outputs to obtain the final global agent queries:

$$q_{agent} = \text{AvgPool}\left(q_{stage}^M\right) \times W_a + b_a, \ q_{agent} \in \mathbb{R}^{r \times c} \tag{6}$$

where $W_a$ and $b_a$ signify the weights and biases, respectively. As a result, we embed semantic information from the segmentation backbone into agent queries and establish an implicit linkage between agent queries and instances within the scene. These agent queries are then guided by the diffusion features to learn contextual relationships between different instances, enabling a more comprehensive understanding of the scene distribution.

### 4.2. Diffusion-Guided Agent Queries Optimization

The diffusion model encapsulates not only prior knowledge of scene distribution but also various visual appearances (e.g., colors and textures), which are irrelevant or even detrimental to perceptual tasks. To guide agent queries in focusing on the scene distribution, we propose a diffusion consistency loss (DCL) that decouples scene distribution knowledge from intricate visual details in diffusion features.

**Extraction of Diffusion Features.** Before that, we need to extract diffusion features. To do this, we first transform the image $x$ to the latent space using the pre-trained encoder $\mathcal{E}$ to obtain the latent code $z_0 = \mathcal{E}(x)$, then use Equation 1 to get the noise samples $z_t$, and finally use the noise predictor $\epsilon_\theta(\cdot)$ to perform the denoising process to get diffusion features:

$$f_d^{(t_s,j)} = \epsilon_\theta^j\left(z_{t_s}, t_s, \mathcal{T}(e)\right), \ f_d^{(t_w,j)} = \epsilon_\theta^j\left(z_{t_w}, t_w, \mathcal{T}(e)\right) \tag{7}$$

where $\mathcal{T}$ is the text encoder, $e$ is the empty text, $1 \leq j \leq K$ and $K$ represents the total number of diffusion features. Different timesteps $t$ correspond to different denoising stages,

where weak noise added at timestep $t_w$ results in diffusion features $f_d^{(t_w,j)}$ with finer-grained details, while strong noise at timestep $t_s$ leads to features $f_d^{(t_s,j)}$ with coarser-grained semantic information.

**The Diffusion Consistency Loss.** Our goal is to enable agent queries to comprehensively understand the scene semantic distribution, guided by diffusion features, to capture the complex contextual relationships between instances within the scenario. To achieve this, we firstly perform a dot product operation on the agent queries $q_{agent}$ with the weak noise diffusion features $f_d^{(t_w,j)}$ to obtain a similarity map:

$$S_j^{t_w} = \text{Softmax}\left(f_d^{(t_w,j)} \times \text{Linear}(q_{agent})^T\right) \tag{8}$$

where Linear is a two-layer MLP with layer normalization. The dot product operation effectively associates the instance semantics of the agent queries with the scene distribution knowledge in the weak-noise diffusion features. As a result, $S_j^{t_w}$ inherently contains the scene distribution prior embedded in the weak-noise diffusion features. However, the visual detail information inherent in the weak-noise diffusion features $f_d^{(t_w,j)}$ inevitably propagates into $S_j^{t_w}$. To mitigate the interference of these details, it is necessary to strip away the potential visual detail within $S_j^{t_w}$. Considering that the primary distinction between strong-noise diffusion features $f_d^{(t_s,j)}$ and weak-noise diffusion features $f_d^{(t_w,j)}$ lies in the low-level visual detail, while both remain essentially identical in representing the scene semantic distribution, we leverage $S_j^{t_w}$ and $q_{agent}$ to reorganize the weak-noise features, with the original strong-noise diffusion features $f_d^{(t_s,j)}$ supervising the reorganization process:

$$\mathcal{L}_{\text{dist}} = \sum_{\hat{h}=1}^{\hat{H}} \sum_{\hat{w}=1}^{\hat{W}} f_d^{(t_s,j)}(\hat{h}, \hat{w}) \log \frac{f_d^{(t_s,j)}(\hat{h}, \hat{w})}{\hat{f}_d^j(\hat{h}, \hat{w})} \tag{9}$$
$$\hat{f}_d^j = S_j^{t_w} \times \text{Linear}(q_{agent})$$

where $\hat{H}$ and $\hat{W}$ represent the height and width dimensions of the feature map $f_d^{(t_s,j)}$, respectively. By enforcing a consistency constraint, visual details in $S_j^{t_w}$ are effectively minimized. This is because if $S_j^{t_w}$ includes excessive visual details, it would result in a significant discrepancy between the reorganized result $\hat{f}_d^j$ and the strong-noise diffusion features $f_d^{(t_s,j)}$. The consistency loss minimizes this discrepancy, ensuring that during the construction of $S_j^{t_w}$, only the semantic distribution of the scene is preserved while visual details are suppressed.

Ultimately, we utilize $S_j^{t_w}$ to optimize the instance semantic representation in the agent queries, resulting in updated feature representations:

$$q_{opt}^j = (f_d^{(t_s,j)})^{\text{T}} \times S_j^{t_w} \tag{10}$$

To guide the agent queries $q_{agent}$ in constructing the scene distribution, we supervise them using $q_{opt}$:

$$\mathcal{L}_{\text{sup}} = \sum_{j=1}^{K} L_\delta(q_{agent}, (q_{opt}^j)^T)$$

$$\text{with } L_\delta(x, y) = \begin{cases} 0.5(x-y)^2 & \text{if } |x-y| < 1 \\ |x-y| - 0.5 & \text{otherwise} \end{cases} \quad (11)$$

**The diffusion consistency loss** is:

$$\mathcal{L}_{\text{diff}} = \mathcal{L}_{\text{sup}} + \mathcal{L}_{\text{dist}} \quad (12)$$

This process enables the agent queries to actively establish contextual relationships between instances within the scene during their generation, fostering a comprehensive understanding of the underlying scene distribution.

### 4.3. Feature Refinement and Mask Prediction

The final step involves leveraging the optimized agent queries to refine the features in the segmentation decoder and predicting the segmentation mask. The typical decoder can be divided into three components: a pixel decoder that extracts features, a transformer decoder that outputs mask embedding, and a mask decoder that outputs class probabilities (Cheng et al., 2022). In our pipeline, we insert the refinement module after the pixel decoder, which takes the multi-scale features output by the pixel decoder and the optimized agent queries as inputs and outputs the refined features. For the refinement module, we use cross-attention and multiple parallel 3×3 depthwise separable convolutions (Chollet, 2017) with different dilation rates to refine feature representations. In particular, we start by feeding the multi-scale features and optimized agent queries into cross-attention to get new feature representations. Next, we concatenate these with the original multi-scale features, perform fusion using depth-separable convolution, and then align dimensions with 1×1 convolution to obtain refined pixel features. The transformer decoder produces a set of mask embeddings, which take the refined pixel features as input. These mask embeddings are then classified by the mask decoder to produce a set of class probabilities. Finally, we use the segmentation loss $\mathcal{L}_{\text{seg}}$ as follows:

$$\mathcal{L}_{\text{seg}} = \lambda_{\text{bce}} \mathcal{L}_{\text{bce}} + \lambda_{\text{dice}} \mathcal{L}_{\text{dice}} + \lambda_{\text{cls}} \mathcal{L}_{\text{cls}} \quad (13)$$

where the predicted mask is optimized using binary cross entropy loss $\mathcal{L}_{\text{bce}}$ and dice loss $\mathcal{L}_{\text{dice}}$, while the predicted classes of the mask embedding are optimized using binary cross entropy loss $\mathcal{L}_{\text{cls}}$. The loss weights $\lambda_{\text{bce}}$, $\lambda_{\text{dice}}$ and $\lambda_{\text{cls}}$ are set to the same values.

**Full Objective.** The full training objective consists of the segmentation loss and the diffusion consistency loss:

$$\mathcal{L}_{\text{total}} = \mathcal{L}_{\text{seg}} + \alpha \mathcal{L}_{\text{diff}} \quad (14)$$

where $\alpha$ controls the weight of the diffusion consistency loss. Notably, during inference, the step involving diffusion features is omitted, ensuring our method remains concise, efficient, and general.

## 5. Experiments

### 5.1. Experimental Setups

**Datasets.** We evaluate the performance of QueryDiff following the domain generalization standard setting (Hümmer et al., 2023; Benigmim et al., 2024; Wei et al., 2024). As synthetic datasets, GTA5 (Richter et al., 2016) provides 24,966 images at a resolution of 1914×1052. As real-world datasets, Cityscapes (Cordts et al., 2016) includes 2,975 images for training and 500 images for validation, with images at 2048×1024 resolution. BDD100K (Yu et al., 2020) consists of 7,000 training images and 1,000 validation images, each at 1280×720 resolution. Mapillary (Neuhold et al., 2017) offers 18,000 training images and 2,000 validation images, with resolutions varying across the dataset. As adverse weather datasets, ACDC (Sakaridis et al., 2021) consists of 406 validated images at 1920×1080 resolution, including night, snow, rain and fog. For simplicity, we abbreviate GTA5, Cityscapes, BDD100K, and Mapillary as G, C, B, and M, respectively; AN, AS, AR and AF denote night, snow, rain and fog subsets of ACDC, respectively.

**Implementation Details.** Following previous works (Benigmim et al., 2024; Wei et al., 2024), we use the decoder from mask2former (Cheng et al., 2022), a widely-used segmentation head compatible with various backbones, including ResNet50 (He et al., 2016), MiT-B5 (Xie et al., 2021), and DINOv2 (Oquab et al., 2023). For the training phase, the AdamW optimizer (Loshchilov & Hutter, 2017) is employed, setting the learning rate at 1e-5 for the backbone and 1e-4 for both the decoder and the learnable queries. We utilize a configuration of 60,000 iterations with a batch size of 4, and crop images to a resolution of $512 \times 512$. We employ Stable Diffusion v2-1 (Rombach et al., 2022) as our diffusion model, which is trained on LAION5B (Schuhmann et al., 2022) and remains frozen throughout.

### 5.2. Comparison with Previous Methods

We comprehensively compare our method with existing DGSS methods. We conduct experiments in three generalization settings: synthetic-to-real (G→C, B, M), real-to-real (C→B, M) and normal-to-adverse (C→AF, AR, AS, AN).

**Synthetic-to-real generalization.** Table 1 presents a comparison between our proposed method and existing DGSS approaches under the G→C, B, M scenario. Compared to the previous SOTA method Rein, our approach achieves a substantial improvement of 2.4 points and consistently surpasses all DGSS methods across various backbone archi-

*Table 1.* Performance comparison between the proposed QueryDiff and existing DGSS methods. Top three results are highlighted as best , second and third , respectively. VFM refers to the vision foundation model. Our results are an average of 3 times.

| Method | Proc. & Year | Backbone | Utilizing Diffusion | synthetic-to-real | | | | real-to-real | | |
|---|---|---|---|---|---|---|---|---|---|---|
| | | | | G→C | G→B | G→M | Avg. | C→B | C→M | Avg. |
| *ResNet based:* | | | | | | | | | | |
| WildNet (Lee et al., 2022) | CVPR2022 | RN101 | ✗ | 44.6 | 38.4 | 46.1 | 43.0 | 50.9 | 58.8 | 54.9 |
| SHADE (Zhao et al., 2022) | ECCV2022 | RN101 | ✗ | 46.7 | 43.7 | 45.5 | 45.3 | 51.0 | 60.7 | 55.8 |
| SAW (Peng et al., 2022) | CVPR2022 | RN50 | ✗ | 39.8 | 37.3 | 41.9 | 39.7 | 53.0 | 59.8 | 56.4 |
| TLDR (Kim et al., 2023) | ICCV2023 | RN101 | ✗ | 47.6 | 44.9 | 48.8 | 47.1 | - | - | - |
| DIDEX (Niemeijer et al., 2024) | WACV2024 | RN101 | ✓ | 52.4 | 40.9 | 49.2 | 47.5 | - | - | - |
| BlindNet (Ahn et al., 2024) | CVPR2024 | RN50 | ✗ | 45.7 | 41.3 | 47.1 | 44.7 | 51.8 | 60.2 | 56.0 |
| FAMix (Fahes et al., 2024) | CVPR2024 | RN101 | ✗ | 49.5 | 46.4 | 52.0 | 49.3 | - | - | - |
| **QueryDiff (Ours)** | - | RN50 | ✓ | **52.5** | **49.2** | **53.3** | **51.7** | **53.7** | **64.1** | **58.9** |
| *Transformer based:* | | | | | | | | | | |
| HRDA (Hoyer et al., 2022) | ECCV2022 | MiT-B5 | ✗ | 57.4 | 49.1 | 61.1 | 55.9 | 58.5 | 68.3 | 63.4 |
| HGFormer (Ding et al., 2023) | CVPR2023 | Swin-T | ✗ | - | - | - | - | 53.4 | 66.9 | 60.2 |
| DGInStyle (Jia et al., 2023) | ECCV2024 | MiT-B5 | ✓ | 58.6 | 52.3 | 62.5 | 57.8 | 58.8 | 68.0 | 63.4 |
| **QueryDiff (Ours)** | - | MiT-B5 | ✓ | **61.9** | **55.4** | **63.2** | **60.2** | **61.9** | **70.7** | **66.3** |
| *VFM based:* | | | | | | | | | | |
| CLOUDS (Benigmim et al., 2024) | CVPR2024 | ConvNext-L | ✓ | 60.2 | 57.4 | 67.0 | 61.5 | - | - | - |
| Rein (Wei et al., 2024) | CVPR2024 | DINOv2-L | ✗ | 66.4 | 60.4 | 66.1 | 64.3 | 63.5 | 74.0 | 68.8 |
| **QueryDiff (Ours)** | - | DINOv2-L | ✓ | **69.1** | **62.3** | **68.6** | **66.7** | **66.1** | **75.9** | **71.0** |

*Table 2.* Normal-to-adverse datasets generalization. Comparison with state-of-the-art methods for DGSS on C → AF, AN, AR, AS. Models are tested on the validation set of ACDC. VFM refers to the vision foundation model. Our results are an average of 3 times.

| Method | Backbone | normal-to-adverse | | | | |
|---|---|---|---|---|---|---|
| | | C→AF | C→AN | C→AR | C→AS | Avg. |
| *ResNet based:* | | | | | | |
| Mask2Former (Cheng et al., 2022) | RN50 | 54.1 | 36.5 | 53.1 | 50.6 | 49.8 |
| HGFormer (Ding et al., 2023) | RN50 | 56.5 | 35.8 | 57.7 | 56.2 | 51.6 |
| **QueryDiff (Ours)** | RN50 | **59.5** | **37.4** | **58.2** | **60.1** | **53.8** |
| *Transformer based:* | | | | | | |
| ISSA (Li et al., 2023) | MiT-B5 | 67.5 | 33.2 | 55.9 | 53.2 | 52.5 |
| Mask2Former (Cheng et al., 2022) | Swin-L | 69.1 | 53.1 | 68.3 | 65.2 | 63.9 |
| HGFormer (Ding et al., 2023) | Swin-L | 69.9 | 52.7 | 72.0 | 68.6 | 65.8 |
| **QueryDiff (Ours)** | MiT-B5 | **73.1** | **53.9** | **72.9** | **70.7** | **67.7** |
| *VFM based:* | | | | | | |
| Rein (Wei et al., 2024) | DINOv2-L | 76.4 | 70.6 | 78.2 | 79.5 | 77.6 |
| **QueryDiff (Ours)** | DINOv2-L | **78.5** | **72.5** | **82.3** | **82.4** | **79.9** |

*Table 3.* Ablation study on primary components. The experiments are conducted using Rein with DINOv2-L backbone, trained on the GTAV dataset. AQ denotes agent queries.

| | AQ | $\mathcal{L}_{sup}$ | $\mathcal{L}_{dist}$ | G→C | G→B | G→M | Avg. | ΔmIoU |
|---|---|---|---|---|---|---|---|---|
| 1 | | | | 66.4 | 60.4 | 66.1 | 64.3 | - |
| 2 | ✓ | | | 68.0 | 60.9 | 67.4 | 65.4 | +1.1 |
| 3 | ✓ | ✓ | | 68.8 | 61.7 | 67.9 | 66.1 | +1.8 |
| 4 | ✓ | ✓ | ✓ | 69.1 | 62.3 | 68.6 | 66.7 | +2.4 |

*Table 4.* Generalization ability test of the proposed QueryDiff across multiple VFMs as backbones. * denotes trainable parameters in backbones.

| Backbone | Training Method | Trainable Params* | mIoU | | | |
|---|---|---|---|---|---|---|
| | | | G→C | G→B | G→M | Avg. |
| CLIP | Full | 304.15M | 51.3 | 47.6 | 54.3 | 51.1 |
| | Freeze | 0.00M | 53.7 | 48.7 | 55.0 | 52.5 |
| | Rein | 2.99M | 57.1 | 54.7 | 60.5 | 57.4 |
| | QueryDiff | 5.73M | 58.9 | 56.0 | 61.9 | 58.9 |
| SAM | Full | 632.18M | 57.6 | 51.7 | 61.5 | 56.9 |
| | Freeze | 0.00M | 57.0 | 47.1 | 58.4 | 54.2 |
| | Rein | 4.51M | 59.6 | 52.0 | 62.1 | 57.9 |
| | QueryDiff | 7.25M | 61.4 | 53.7 | 64.2 | 59.8 |
| DINOV2 | Full | 304.20M | 63.7 | 57.4 | 64.2 | 61.8 |
| | Freeze | 0.00M | 63.3 | 56.1 | 63.9 | 61.1 |
| | Rein | 2.99M | 66.4 | 60.4 | 66.1 | 64.3 |
| | QueryDiff | 5.73M | 69.1 | 62.3 | 68.6 | 66.7 |

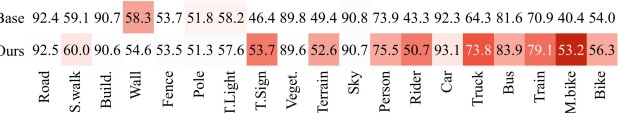

| | Road | S.walk | Build. | Wall | Fence | Pole | T.Light | T.Sign | Veget. | Terrain | Sky | Person | Rider | Car | Truck | Bus | Train | M.bike | Bike |
|---|---|---|---|---|---|---|---|---|---|---|---|---|---|---|---|---|---|---|---|
| Base | 92.4 | 59.1 | 90.7 | 58.3 | 53.7 | 51.8 | 58.2 | 46.4 | 89.8 | 49.4 | 90.8 | 73.9 | 43.3 | 92.3 | 64.3 | 81.6 | 70.9 | 40.4 | 54.0 |
| Ours | 92.5 | 60.0 | 90.6 | 54.6 | 53.5 | 51.3 | 57.6 | 53.7 | 89.6 | 52.6 | 90.7 | 75.5 | 50.7 | 93.1 | 73.8 | 83.9 | 79.1 | 53.2 | 56.3 |

*Figure 3.* Comparison of the class-wise IoU on GTA → Cityscapes using Rein with and without our method. The red colors visualize the differences in class-wise IoU between the baseline and our method.

tectures, including ResNet, MiT-B5, and DINOV2. In particular, when using ResNet50, our method achieves 51.7%, i.e. +7% compared to BlindNet, which also uses ResNet50, and +2.4% compared to FAMix using the stronger ResNet101.

**Real-to-real generalization.** Under this evaluation setup, models are trained on Cityscapes and tested on BDD100K and Mapillary. The results, starting from the ninth column of Table 1, confirm that our method consistently delivers better domain generalization performance across various datasets and backbone configurations, underscoring its effectiveness in adapting to diverse real-world scenarios.

**Normal-to-adverse generalization.** Under this evaluation setup, all the models are trained on Cityscapes and tested on ACDC. Table 2 highlights that our method achieves consistently better performance than competing approaches across all three backbone variants. When using ResNet50, our method improves performance from 51.6 to 53.8 compared with HGFormer. As for the backbone of MiT-B5 and DINOV2-L, our method respectively improves the mIoU by 1.9 and 1.6 compared with the previous SOTA method.

**Comparison of the class-wise IoU.** We gauge the impact

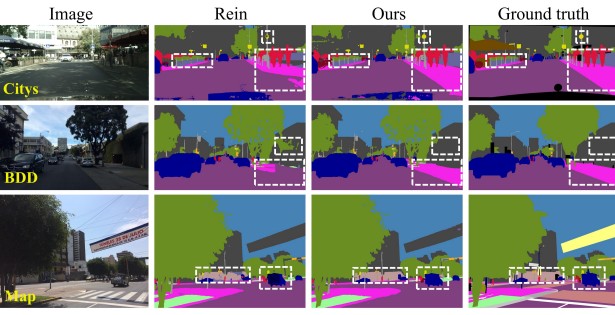

*Figure 4.* Qualitative comparison under G→C, B, M generalization setting. From left to right: target image, the visual results predicted by Rein, Ours, and Ground Truth. We deploy the white dash boxes to highlight different prediction parts.

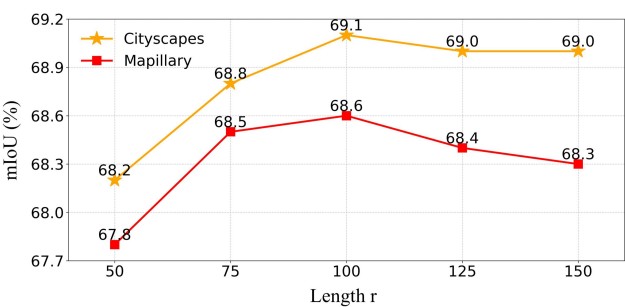

*Figure 5.* Ablation study on agent queries length $r$.

of our method on class-wise IoU scores, as shown in Figure 3, which demonstrates notable performance gains across multiple classes, especially in rarer ones like rider, truck, train, and motorbike. The heatmap affirms the capability of our method across a wide range of classes.

**Qualitative results.** We visually compare the segmentation results with the previous SOTA Rein under G→C, B, M setting in Figure 4. The results highlighted by white dash boxes show that our method creates more reasonable class predictions as well as more complete spatial distributions. We think it is because the proposed method learns the correct knowledge of the scenario, which is crucial for improving the generalization of the model.

### 5.3. Ablation Studies

In this section, we present comprehensive experiments to validate the effectiveness of our method.

**Effect of components.** We evaluate the effectiveness of the primary components under G→C, B, M settings, focusing on agent queries, $\mathcal{L}_{\text{sup}}$ and $\mathcal{L}_{\text{dist}}$ in the proposed method. The baseline model is based on Rein. As shown in Table 3, we can observe: (1) Each module positively contributes to the model's performance, demonstrating the individual effectiveness of these components. (2) The combination of all three modules yields the highest mIoU of 66.7 (+2.4

*Table 5.* Ablation study on different sets of timesteps.

| $t_w$ | $t_s$ | C | B | M | Avg. |
|---|---|---|---|---|---|
| 0 | 50 | 68.2 | 62.0 | 68.4 | 66.2 |
| 50 | 100 | 68.7 | 62.1 | 68.6 | 66.5 |
| 0 | 100 | **69.1** | **62.3** | **68.6** | **66.7** |
| 75 | 100 | 68.3 | 62.0 | 68.2 | 66.1 |
| 100 | 200 | 68.5 | 62.1 | 67.1 | 65.9 |
| 0 | 200 | 67.6 | 61.9 | 67.4 | 65.6 |

*Table 6.* Quantitative comparison of different diffusion models.

| Model | Rein | +SD 1.4 | +SD 1.5 | +SD 2.1 |
|---|---|---|---|---|
| mIoU | 64.3 | **66.4** | **66.6** | **66.7** |

over the baseline), highlighting their synergy in enhancing cross-domain segmentation.

**Investigating various VFMs.** We evaluate QueryDiff across multiple Vision Foundation Models (VFMs), including CLIP (Radford et al., 2021), SAM (Kirillov et al., 2023), and DINOv2 (Oquab et al., 2023), under full fine-tuning, lightweight fine-tuning (Rein), and frozen backbone schemes (Wei et al., 2024) in the G → C, B, M setting. As shown in Table 4, QueryDiff achieves the highest mIoU, significantly outperforming other methods.

**Study on agent queries length $r$.** The core component of our method is a set of agent queries. We explore various lengths for the agent query sequence, ranging from 50 to 150. As demonstrated in Figure 5, models with $r = 100$ achieve a strong mIoU of 69.1% and 68.6% on Cityscapes and Mapillary datasets, respectively.

**The choice of $t_w$ and $t_s$.** Noise intensity increases with the timestep value, with higher values introducing stronger perturbations. Previous studies (Xu et al., 2023; Baranchuk et al., 2022) reveal that semantically meaningful diffusion features are primarily generated within the range of 0 to 200, and that neighboring timesteps often produce highly similar representations. As shown in Table 5, we empirically adapt $t_w = 0$ and $t_s = 100$ to ensure that agent queries can capture high-dimensional semantics while avoiding being influenced by trivial information.

**Comparing different stable diffusion.** In Table 6, we use Rein as the baseline model and conduct the comparative experiment with three currently dominant stable diffusion models. The final results are not sensitive to the choice of different diffusion models, and our proposed method, when combined with different diffusion models, consistently outperforms the baseline model.

## 6. Conclusion

In this work, we propose QueryDiff, a novel agent query-driven learning framework based on diffusion model guidance for DGSS. QueryDiff leverages agent queries as an interface to mine scene distribution priors embedded in

diffusion model. Also, to avoid the interference of visual details, we propose diffusion consistency loss to enable agent queries to focus on domain-invariant semantic information. Extensive experiments demonstrate that QueryDiff significantly surpasses previous SOTA methods across various benchmarks, highlighting its effectiveness. This work not only bridges a critical research gap but also establishes a new standard for domain generalized semantic segmentation.

## Impact Statement

This paper presents work whose goal is to advance the field of Machine Learning. There are many potential societal consequences of our work, none which we feel must be specifically highlighted here.

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
