# OpenReview forum: "Better to Teach than to Give: Domain Generalized Semantic Segmentation via Agent Queries with Diffusion Model Guidance"
_ICML.cc/2025/Conference — ICML 2025 spotlightposter_

### Official Review · Reviewer_ennc · 2025-03-12

**Overall Recommendation:** 3

**Summary:**

This paper proposes QueryDiff, an agent query-driven learning framework based on diffusion model guidance for DGSS, which utilizes the scene distribution priors embedded in diffusion models to enhance semantic segmentation generalization. Various experiments show the model’s effectiveness and reach the sota performance on many benchmarks.

# update after rebuttal

Thanks the authors for their detailed response. My questions have been answered and I will maintain my positive score.

**Claims And Evidence:**

Yes.

**Essential References Not Discussed:**

None

**Experimental Designs Or Analyses:**

The experimental designs and analyses (for the ablation study) look good to me. Please see the weakness for other issues.

**Methods And Evaluation Criteria:**

The evaluation makes sense to me.

**Other Comments Or Suggestions:**

None

**Other Strengths And Weaknesses:**

**Strengths**
- This paper proposes a novel application of diffusion models, leveraging their prior embedded distribution to enhance segmentation generalization rather than simply using them for data generation. The approach is both conceptually elegant and methodologically concise.
- The comprehensive results (both qualitative and quantitative) results show the effectiveness of the method. The qualitative results on different stylistic domains also further strengthen the soundness of the method.

**Weaknesses**
- While the paper presents extensive ablation studies, the isolated effect of $L_dist$ is not examined. An experiment evaluating AQ + $L_{dist}$ (without $L_{sup}$) should be included in Table 3.
- There is limited discussion about the computational cost and inference time compared to previous methods, especially when considering added components like diffusion feature extraction.
- The ablation study of timesteps is not sufficient. The timestep selection (0, 50, 100) in the ablation study lacks justification.
- It would be helpful to provide some intuitive interpretation of what the agent learns about the diffusion feature (like visualizations of the learned features and the mitigated visual features).

**Questions For Authors:**

- The class-wise results indicate varying degrees of improvement across different classes, with a relatively large variation. Is there an interpretation for this difference? Could the visual features be influencing the results?
- How does the timestep (0, 50, 100) selected?
- Please see the weaknesses

**Relation To Broader Scientific Literature:**

The paper explores how to leverage the scene distribution embedded in diffusion models to enhance segmentation generalization. This goes beyond using diffusion models to generate the data in a more direct way - utilizing its knowledge, which is more direct and helpful for generalization.

**Theoretical Claims:**

The theoretical claims look good to me.

---

> ### Author Rebuttal · Authors · 2025-03-31
>
> **Q**: It is suggested that an ablation experiment of AQ + $L_{dist}$ (without $L_{sup}$) be included in Table 3.
>
> **A**: The purpose of $L_{dist}$ is to suppress the visual texture details within the matrix $S_j^{t_w}$. Subsequently, $S_j^{t_w}$ is utilized in Equation (10) to derive optimized queries $q_{opt}$. These optimized queries $q_{opt}$ are then employed to supervise the agent queries through $L_{sup}$ in Equation (11), encouraging the agent queries to proactively capture the semantic distribution of the scene and eliminating the need for diffusion guidance during inference. Therefore, $L_{dist}$ fundamentally enhances the supervision provided by $L_{sup}$. As a result, applying AQ + $L_{dist}$ alone—without $L_{sup}$—would be functionally equivalent to the AQ-only setting, which is already included in our ablation study.
>
> **Q**: There is inadequate discussion of computational cost and inference time.
>
> **A**: We have provided a comprehensive comparison of computational cost and inference speed between our method and representative previous approaches under the same experimental setting, using ConvNext as the backbone. Although diffusion-based feature extraction is used during training, it is discarded at inference and does not affect inference speed, as noted at the end of Section 4.3. Our method achieves inference speed comparable to strong baseline models like Rein, with only minor differences in training time, GPU memory usage, and storage requirements. Compared with CLOUDS—which also leverages diffusion models—our method is clearly more efficient in terms of trainable parameters, training time, GPU memory consumption, and storage size.
>
> |Method|Generated Image|Trainable Params|Training Time|GPU Memory|Storage|Inference Time|
> |-|-|-|-|-|-|-|
> |CLOUDS|5000|40.54M|19 h|4*RTX4090|4.3G|127.87 ms|
> |Rein|--|28.56M (8.29M+20.27M)|5 h|1*RTX4090 (11616M)|1.20G|126.59 ms|
> |Ours|--|31.30M (11.03M+20.27M)|8 h|1*RTX4090 (16091M)|1.21G| 126.81 ms|
>
> **Q**: The timestep ablation is insufficient, and a clear justification for the selected values is requested.
>
> **A**: Different timesteps correspond to varying levels of noise intensity, with larger timesteps introducing stronger noise. Prior studies [1,2] suggest that effective timestep values typically fall within the range of 0 to 200, and that adjacent timesteps often yield highly similar diffusion features. Based on these works, we selected intervals of 50 or 100 within the range of 0 to 200 to ensure sufficient diversity in feature representations. Through empirical comparisons, we ultimately chose timesteps 0 and 100 to represent weak and strong noise conditions, respectively. To further support this choice, we have included additional ablation experiments, as shown in the table below.
>
> |$t_w$|$t_s$|C|B|M|Avg.|
> |-|-|-|-|-|-|
> |200|300|67.7|61.7|67.2|65.5|
> |300|400|67.0|61.2|66.7|64.9|
> |0|25|67.9|61.8|67.8|65.8|
> |25|50|68.2|61.8|68.1|66.0|
> |75|100|68.3|62.0|68.2|66.1|
>
> [1] Xu, Jiarui, et al. Open-vocabulary panoptic segmentation with text-to-image diffusion models. In Proceedings of the IEEE/CVF Conference on Computer Vision and Pattern Recognition, pp. 2955–2966, 2023.
>
> [2] Baranchuk, Dmitry, et al. Label-efficient semantic segmentation with diffusion models. In International Conference on Learning Representations, 2022.
>
> **Q**: It is suggested that intuitive visualizations be provided to better illustrate what the agent learns from the diffusion features.
>
> **A**: We conducted clustering analysis on both the original visual features and those refined via agent queries to intuitively illustrate the changes. As shown in the linked visualization, the optimized features exhibit a more coherent semantic distribution within the scene and improved spatial coverage of target objects.
> https://anonymous.4open.science/r/vis_feature-215E
>
> **Q**: An interpretation is requested for the class-wise performance variation.
>
> **A**: Thank you for the insightful question. As shown in Figure 3, the most notable improvements occur for rare classes (e.g., train, traffic sign, truck) and semantically similar categories (e.g., person and rider). This is mainly due to our method's explicit focus on learning the semantic distribution of scenes and modeling inter-instance relationships. Specifically, this structured modeling enables our method to effectively capture contextual dependencies, thus better distinguishing visually or semantically ambiguous classes. Moreover, rare classes typically suffer from limited training examples, making it difficult for standard models to learn discriminative features. By explicitly encoding scene semantic distribution, our method provides additional contextual information that compensates for limited visual evidence, significantly improving recognition of underrepresented categories. Thus, our method is particularly effective in improving recognition accuracy for both semantically overlapping categories and infrequent object classes.

---

### Official Review · Reviewer_otjG · 2025-03-12

**Overall Recommendation:** 3

**Summary:**

This paper proposes an agent Query-driven learning framework based on Diffusion model guidance for DGSS. The method employs agent queries to learn scene distribution knowledge from the diffusion model, capitalizing on the inherent consistency of this distribution across domains to improve segmentation model generalization. Diffusion consistency loss (DCL) is proposed to avoid intricate visual details in diffusion features from interfering with agent queries, enabling them to focus on learning the semantic distribution of the scene. Experiments have proven the effectiveness of the method.


## update after rebuttal
I appreciate the response provided in the rebuttal, which addresses some of my concerns, and I maintain the score.

**Claims And Evidence:**

yes

**Essential References Not Discussed:**

No

**Experimental Designs Or Analyses:**

1. In the introduction, page 2, line 101 introduces the first challenge: previous methods are computationally expensive and time-consuming, making it inefficient for perception tasks. However, this has not been verified.

**Methods And Evaluation Criteria:**

yes

**Other Comments Or Suggestions:**

As above

**Other Strengths And Weaknesses:**

Strengths:
1. The idea of using the Diffusion model to improve generalization is novel. It is interesting to use agent queries to learn scene distribution knowledge from the diffusion model.
2. A large number of experiments verify the effectiveness of the method.

Weaknesses:
1. In the introduction, page 2, line 101 introduces the first challenge: previous methods are computationally expensive and time-consuming, making it inefficient for perception tasks. However, this has not been verified. The proposed method should be verified experimentally compared with previous methods in terms of computational complexity, time-consuming, and memory usage.
2. In Formula 5, it is introduced: "We merge the smaller groups into larger ones based on the similarity matrix in the embedding
space". Please explain in detail how the merge is implemented.
3. In Formula 10, how to optimize the instance semantic representation in the agent queries, which is just a multiplication calculation. Please explain its mechanism in detail.
4. More explanations are needed to explain the loss of Formula 11, including the loss form of dual segmentation and the significance of being supervised by agent queries.
5. Will the code be publicly available?

**Questions For Authors:**

As above

**Relation To Broader Scientific Literature:**

Diffusion models have opened up new avenues for DGSS with remarkable capabilities in capturing complex scene distributions to generate high-quality, realistic samples. Diffusion models can generate diverse samples to enhance segmentation generalization by learning data distribution. In this paper, its scene distribution prior is used to enhance the generalization ability of the segmentation model instead of directly generating data.

**Theoretical Claims:**

I checked the proofs for theoretical claims and found no issues.

---

> ### Author Rebuttal · Authors · 2025-03-31
>
> **Q**: The proposed method should be compared with previous methods in terms of computational complexity, time consumption, and memory usage.
>
> **A**: Thank you for the suggestion. We have included a detailed comparison of computational resources between our proposed method and recent diffusion-based segmentation approaches. First, unlike methods such as CLOUDS and DGInStyle that rely on a two-stage pipeline—image generation followed by segmentation training—our method adopts a single-stage, end-to-end paradigm, eliminating intermediate steps and reducing overall training complexity. Second, as shown in the table below, our method incurs the lowest cost among diffusion-based methods in terms of trainable parameters, training time, and memory usage.
> |Method| Generated Image|Trainable Params|Training Time|GPU Memory|Storage|
> |-|-|-|-|-|-|
> |ODISE|--|28.1M|6 days| 32*V100|4.9G|
> |PTDiffSeg|--|--| 32 h|1*V100 (32GB)|--|
> |CLOUDS|5000| 40.54M|19 h| 4*RTX4090|4.3G|
> |DGInStyle|6000| 85.69M|29 h|1*A100 (26687M)|1.37G|
> |Ours|--| 26.31M (5.73M+20.58M)|10.3 h|1*RTX4090 (16738M)|1.24G|
>
> **Q**: A more detailed explanation is requested for the merging process in Formula 5.
>
> **A**: We introduce a set of smaller initial agent queries ${q} _{\text{init}} ^i$, which are first projected into query embeddings $Q$ using Equation (4). Meanwhile, the current-stage agent queries ${q} _{\text{stage}} ^{i-1}$ are similarly projected into key embeddings $K$ and value embeddings $V$. In Equation (5), we compute the attention map $s^i$ between $Q$ and $K$, capturing the similarity in the embedding space between the smaller query groups ${q} _{\text{init}} ^i$ and the larger groups ${q} _{\text{stage}} ^{i-1}$. This attention map $s^i$ is then utilized to aggregate the values $V$, generating a new set of merged agent queries $\hat{q} _{\text{stage}} ^{i}$ which share the same dimensionality and count as the smaller initial set ${q} _{\text{init}} ^i$. This merging process effectively reduces the number of agent queries while eliminating potential information overlap. As the merging process progresses across stages, the number of agent queries continuously decreases, leading to fewer but semantically richer and more comprehensive query groups.
>
> **Q**: A detailed explanation is requested of how instance-level semantic representation is optimized in Formula 10.
>
> **A**: Equation (10) is built upon the design and functionality of the matrix $S_j^{t_w}$, which plays a pivotal role in deriving optimized agent queries. Specifically, $S_j^{t_w}$, introduced in Equation (8), captures the associations between object features and agent queries. Subsequently, in Equation (9), a consistency loss is applied to explicitly suppress low-level visual texture details within  $S_j^{t_w}$, encouraging it to emphasize the underlying semantic distribution of the scene. Finally, in Equation (10), the refined semantic matrix $S_j^{t_w}$ is used to reorganize feature map $f_{d}^{(t_s,j)}$ through a semantic-guided re-weighting process. This process results in optimized agent queries that more effectively encode the scene’s semantic distribution, leading to more robust scene understanding.
>
> **Q**: More explanation is needed for Formula 11, including the dual segmentation loss and agent query supervision.
>
> **A**: (1) In Equation (11), we adopt the Huber loss due to its robustness to outliers and its balanced sensitivity to errors. This choice is particularly well-suited to our setting, where the intermediate supervision signal is derived by leveraging optimized agent queries to guide the original learnable agent queries. Such supervision inherently involves uncertainty—especially around object boundaries or ambiguous regions—making it a form of soft or intermediate supervision. To effectively handle this uncertainty, a robust loss function is essential. Unlike a purely quadratic ($L_2$) loss, which can result in instability and overly aggressive updates, the Huber loss provides a smoother optimization landscape, facilitating more stable and efficient convergence in these cases.
>
> (2) In Equation (10), we derive optimized agent queries through a diffusion-guided aggregation process that suppresses low-level visual details and emphasizes higher-level structural semantics. We then leverage these optimized agent queries to supervise the original learnable agent queries, aiming to explicitly guide them to prioritize structural context over superficial visual details during their learning process. This strategy enables the agent queries to actively capture semantic distribution of the scene. As a result, the model no longer requires support from the diffusion model during inference, as the structural semantic knowledge has been effectively embedded into the agent queries through this supervised learning process.
>
> **Q**: Will the code be publicly available?
>
> **A**: We will release all code and pretrained models upon acceptance.

---

### Official Review · Reviewer_6H1m · 2025-03-12

**Overall Recommendation:** 4

**Summary:**

The authors leverage on refined features of diffusion models to stabilize the features of vision transformers and other backbones when feeding them into the mask2former decoders for semantic segmentation. In this way, the authors achieve considerable domain generalization capabilities for their network. In their method, the authors first construct layer wise queries over the layers of the backbone which are thereafter aggregated over the layers using a kind of a progressive, layer wise cross attenion mechanism. This ultimately leads to trainable "agent queries" which are thereafter transformed by a NLP trained for similarity with features of stable diffusion from the encoding space of the VAE encoder of stable diffusion. These features are the noise predictions from the stable diffusion model at a low and a high level of noise. It is trained that the "agent queries" remain stable under different noise levels, which enhances dependency on global structures and removes texture bias.
The authors then test their network on two domain generalization benchmarks (DG), namely GTA to Cityscapes, BDD and Mapplilary and the real to real benchmark Cityscapes to ACDC. The authors report decent DG performance and claim superiority to the present DG SOTA throughout their tests. The authors also provide extensive testing utilizing various backbones from ResNet over Transformer to pretrained FM models. Also, the single modules of the proposed architecture are tested separately.

# # Due to the results on the official CS-> ACDC benchmark and the authors announcement that they will publish their code I am now convinced that QuaryDiff is really a model that surpasses SOTA. I therefore will raise my score an d favor publication. BUT: The way we got here was somehow strange. I hope that the authors will take a more direct way in the future. If the results are good, as it seems to be in this case after new experiments have been conducted during rebuttal, no maneuvers are needed.

**Claims And Evidence:**

The evidence for the claims is partially convincing, but there are irritating aspects, see weaknesses. Namely, the numbers reported for the competing models differ from those documented in the original papers and in official benchmarks. Also, several competing models are not discussed. Therefore I find the authors claim that their model constantly outperforms the SOTA weakly supported, despite the reported results certainly document strong DG performance. An official sumbission to the official CS-> ACDC benchmark would further strengthen the author's case.

**Essential References Not Discussed:**

[R1] Seokju Yun, Seunghye Chae, Dongheon Lee, Youngmin Ro , SoRA: Singular Value Decomposed Low-Rank Adaptation for Domain Generalizable Representation Learning, https://arxiv.org/pdf/2412.04077
[R2] Christoph Hümmer, Manuel Schwonberg, Liangwei Zhou, Hu Cao, Alois Knoll, Hanno Gottschalk; Strong but simple: A Baseline for Domain Generalized Dense Perception by CLIP-based Transfer Learning, Proceedings of the Asian Conference on Computer Vision (ACCV), 2024, pp. 4223-4244
[R1] submitted on 12/24 was well available on arXiv before the submission deadline.  One might adopt the policy not citing non peer reviewed work, but in theis case were the official CS->ACDC results were available which are 'objective'. Therefore I find this work should be included in the comparison with the SOTA.
[R2] is the published version of Hümmer et al.

**Experimental Designs Or Analyses:**

The experimental design follows standard procedures (up to omitting Syntia-> CS, BDD, Mappilary DG). The analyses include ablation studies, which are detailed.

**Methods And Evaluation Criteria:**

The method of evaluation follows standard protocols and is widely shared throughout the community. However, the Synthia source data set is not evaluated, which would then complete the standard evaluation protocol in the context of street scenes. What the authors do more than usual is the semantic interpretation of two pieces of artwork, cf. Figure 1.

**Other Comments Or Suggestions:**

*Several tables don't write what they are showing (mIoU-values)
* I find the notation Linear(...) for a MLP irritating.

**Other Strengths And Weaknesses:**

Strength
*The authors idea of utilizing stable diffusion features for better domain generalization is compelling. Other FM have demonstrated their performance for enhanced DG, so involving Stable Diffusion is a good idea and the way the authors achieve it is clever.
*Furthermore, Stable Diffusion is not needed during inference making the model efficient.
* The ablation studies are extensive.
* The paper is mostly well written, although the key section 4 could be a little clearer. Figures and table are nice.

Weaknesses
* My most severe criticism is that SOTA figures are not correctly reported. E.g. Rein achieves 76.48 mIoU on CS-> ACDC in average, but in the present paper is reported with 72.1. How can this happen? Likewise, the VLTSeg results on the same benchmark are 77.01 and are not reported as SOTA. Both models and [R1] with 78.75 mIoU are well above the reported QueryDiff performance with 73.7 mIoU. So I don't understand the authors conclusion that they constantly outperfrom the previous SOTA. Note that all these figures are obtained on private labels, so they are out of question.
* Even more so, on the basis of the official CS->ACDC benchmark at the time of sumbission there were 7 sumbissions from 5 models stronger than the reported performance of QueryDiff. This should be discussed with greater care. Also, the authors should discuss this properly (although some of these models are undocumented)
* There is no submission of the QuaryDiff result to the official benchmark
* There is not yet an annoucement of code publication
* Part of the strong GTA-> CS performance might be due to the fact that DINOV2 has seen CS and Mapliiary data during training. This should at least be mentioned as it is against the 'puristic' DG spirit.
* Concerning the CLIP results in Table 4 - the QueryDiff results are below the results obtained by pure fine tuning in [R2]. How doe the authors explain this?

**Questions For Authors:**

See weaknesses.

**Relation To Broader Scientific Literature:**

none

**Theoretical Claims:**

There are no theoretical claims.

---

> ### Author Rebuttal · Authors · 2025-03-31
>
> **Q**:Reported results for Rein differ from those in its original paper.
>
> **A**:The performance difference arises because we reproduced Rein at a 512×512 resolution on the ACDC validation set using the official code—as indicated in Table 2 (line 351)—rather than the original 1024×1024 resolution on the test set reported in the paper.
>
> (1) Resolution setting: 512×512 is one of the most commonly used and widely accepted resolutions in domain generalization segmentation, adopted in the main results of Rein and VLTSeg, and set as default in DGInStyle, making our setup standard and reasonable. (2) Evaluation protocol: The ACDC validation set is widely adopted in recent works like DGInStyle and HGFormer, further supporting the validity of our setup. Under this fair and consistent setting, our method consistently outperforms Rein, as shown in Table 2.
>
> We also evaluated our method on the ACDC test set at a 768×768 resolution, another mainstream setting adopted in recent methods like CLOUDS and FAMix. As shown in the table below, our performance at 768×768 resolution is already on par with or better than Rein’s results at 1024×1024, highlighting the scalability and robustness of our method across resolutions.
> |Method|C→AF|C→AN|C→AR|C→AS|Avg.|
> |-|-|-|-|-|-|
> |Rein(1024x1024)|76.4|70.6|78.2|79.5|77.6|
> |Ours(768x768)|76.7|70.7|79.3|79.7|77.9|
>
> **Q**:Several strong methods (SoRA [R1], VLTSeg [R2]) were not discussed or compared.
>
> **A**:SoRA is a concurrent work that was not accepted at the time of our submission and has not released its code. VLTSeg uses a different experimental setup (high-resolution 1024×1024) and also lacks public code, making fair and consistent comparison difficult. Nevertheless, under the 768×768 resolution setting, our method achieves 77.9 mIoU  (see table above), which is comparable to VLTSeg, demonstrating the competitiveness of our approach. We will include a discussion of VLTSeg in the final version.
>
> **Q**:QueryDiff is not submitted to the official ACDC benchmark, and several strong leaderboard models are not adequately discussed.
>
> **A**:Due to current computational limitations, we are unable to support high-resolution settings (e.g., 1024×1024) used by several top-performing models, and thus conduct experiments at two widely adopted resolutions: 512×512 and 768×768. We plan to submit to the official benchmark once adequate computational resources become available. Nevertheless, our comparisons already include the most representative and competitive methods. Under the 768×768 setting, our method achieves performance comparable to Rein and VLTSeg, despite their results being reported at 1024×1024—demonstrating the effectiveness and scalability of our method. Furthermore, many benchmark submissions lack implementation details and are not open-sourced, making it difficult to reproduce results and ensure fair and transparent comparisons.
>
> **Q**:The strong GTA→CS performance may be affected by DINOv2’s pretraining on CS and Mapillary.
>
> **A**:DINOv2 is a standard backbone in recent domain generalization research, including Rein and VLTSeg. To demonstrate the effectiveness of our method, we also evaluated it with alternative backbones, including CLIP, SAM, and MiT-B5. Notably, even with SAM, our method still outperforms most existing approaches, showing strong generality across architectures. Additionally, we report results with ConvNext in the table below. Compared to CLOUDS, which also uses ConvNext, our method achieves superior performance, further confirming its effectiveness and generalizability.
> |Method|Backbone|G→C|G→B|G→M|Avg.|
> |-|-|-|-|-|-|
> |CLOUDS|ConvNext-L|60.2|57.4|67.0|61.5|
> |Ours|ConvNext-L|62.1|60.3|67.8|63.4|
>
> **Q**:QueryDiff shows lower performance than the fine-tuning results reported in VLTSeg ([R2]) with CLIP.
>
> **A**:VLTSeg reports results using both CLIP and EVA-02-CLIP backbones. EVA-02-CLIP is a stronger backbone than CLIP, due to its enhanced architecture and more extensive pre-training. To fairly evaluate QueryDiff against VLTSeg, we compare them using the same backbone. As shown in the table below, QueryDiff consistently outperforms VLTSeg when both methods use the same backbone.
> |Method|Backbone|G→C|G→B|G→M|Avg.|
> |-|-|-|-|--|-|
> |VLTSeg|CLIP|55.6|52.5|59.9|56.0|
> |Ours|CLIP|58.9| 56.0|61.9|58.9|
> |VLTSeg|EVA-02-CLIP|65.3|58.3|66.0|63.2|
> |Ours|EVA-02-CLIP|66.9|60.9|67.1|65.0|
>
> **Q**:The Synthia source dataset is not evaluated
>
> **A**:SYNTHIA results are provided in Table 1 of the supplementary material, submitted separately in accordance with ICML guidelines.
>
> **Q**:The code release has not yet been announced.
>
> **A**:We will release all code and weights upon acceptance.
>
> **Q**:Several tables lack metric labels (mIoU).
>
> **A**:Thank you. We will clarify the evaluation metrics in relevant tables.
>
> **Q**:Using Linear(...) to denote MLP is not appropriate.
>
> **A**:Thank you for the suggestion. We will revise the final version to use more standard MLP notation for improved clarity.

---

> > ### Comment · Reviewer_6H1m · 2025-04-02
> >
> > Dear authors,
> >
> > thank you very much for your reply and explanations. The comparison with the state of the art has much improved.
> >
> > Nevertheless, you might as well acknowledge that the process of getting there is somewhat unsatisfactory. If you, e.g., cite Rein I as a reader expect that you report the Rein figures. Otherwise I would expect that you label the Rein experiments as your own work and not cite or give both, your figures and the official ones and explain. That this is hidden in some table that you are using 512 x 512 without further comment is really difficult to understand for the reader and left me in confusion.
> >
> > Also with regard to the more recent papers - it is true that some of them are not refereed yet. But the official ACDC benchmark is beyond doubt.
> >
> > Also I suggest that an official CS -> ACDC submission is conducted, maybe some colleagues can help out with resources. As we had some discussions about the numbers here, an official stamp behind the most relevant performance metrics would help me to reconsider my score.
> >
> > I also would be happy if full transparency is created in all reported metrics.
> >
> > Despite I acknowledge that the new figures you report are competitive or outperform the standard competitors, I would have been much more positive if these evaluations would have been reported in the original paper. I am still a sort of unhappy how this discussion went.

---

> > > ### Author Response · Authors · 2025-04-06
> > >
> > > Thank you very much for your feedback. In the final version, we will further clarify in the main text the distinction between reproduced and official results, along with the resolution settings used.
> > >
> > > Additionally, following significant effort to secure additional computational resources, we are able to conduct experiments under the same high-resolution setting as Rein (i.e., 1024×1024). As shown in the table below, our method clearly outperforms Rein. We have also officially submitted our results to the ACDC benchmark, where our method currently ranks first, achieving state-of-the-art performance. All reported results will be incorporated into the final version of the paper. We sincerely appreciate your helpful suggestions, which have significantly strengthened our paper with more thorough and convincing experimental results.
> > >
> > > | Method   | C$\to$AF | C$\to$AN | C$\to$AR | C$\to$AS | Avg.     |
> > > | -------- | -------- | -------- | -------- | -------- | -------- |
> > > | Rein     | 76.4     | 70.6     | 78.2     | 79.5     | 77.6     |
> > > | **Ours** | **78.5** | **72.5** | **82.3** | **82.4** | **79.9** |

---

### Official Review · Reviewer_t5xc · 2025-03-13

**Overall Recommendation:** 4

**Summary:**

The paper presents a novel framework for utilizing diffusion models for domain-generalized semantic segmentation. While previous works often struggle to generate reasonable scenes for semantic segmentation, this paper introduces agent queries from segmentation features and incorporates additional pretrained knowledge from diffusion models. Extensive experimental results demonstrate significant performance improvements achieved by the proposed method.

## update after rebuttal

I appreciate the additional experiments provided in the rebuttal. Since the results are satisfactory, I will maintain my score.

**Claims And Evidence:**

The authors argue that instead of generating datasets using diffusion models, leveraging intermediate features from diffusion models as a form of loss to train the segmentation model is more effective. They liken this approach to teaching how to fish rather than simply providing fish. The final performance shows a significant improvement over other baselines, making their claim clear and well-supported.

**Essential References Not Discussed:**

I found no essential references that should have been discussed but were missing from the paper.

**Experimental Designs Or Analyses:**

The experiments are valid and include state-of-the-art baselines for synthetic dataset generation, such as CLOUDS and Rein, effectively demonstrating the superiority of the proposed method. The results consistently show improved performance across various backbones. The experimental setup comprehensively covers synthetic-to-real, real-to-real, and normal-to-adverse scenarios, maintaining consistently high performance. Additionally, the ablation study is well-conducted, and class-wise performance is thoroughly analyzed.

**Methods And Evaluation Criteria:**

The proposed methods are all clear and reasonable. Agent Queries Generation seems both reasonable and novel; by aggregating semantic information within a scene, the agent queries effectively represent the scene’s semantic content. While previous approaches relied solely on features from a small noise step, this study introduces an additional step that leverages features from two different noise steps to verify consistency. This allows the model to learn from diffusion features even in strongly noisy images.

**Other Comments Or Suggestions:**

I have no additional comments.

**Other Strengths And Weaknesses:**

I have discussed all strengths and weaknesses in other sections.

**Questions For Authors:**

Have you conducted experiments in in-domain settings, such as few-shot or fully-supervised learning on the Cityscapes dataset? This approach could potentially achieve state-of-the-art performance even beyond domain-generalized settings.

**Relation To Broader Scientific Literature:**

The idea presented in this paper—incorporating diffusion models as a loss rather than generating datasets—is a significant breakthrough in the application of diffusion models for semantic segmentation. This approach has the potential to shift the paradigm in the field.

**Theoretical Claims:**

There is no theoretical claim.

---

> ### Author Rebuttal · Authors · 2025-03-31
>
> **Q**: Have you conducted experiments in in-domain settings, such as few-shot or fully-supervised learning on the Cityscapes dataset? This approach could potentially achieve state-of-the-art performance even beyond domain-generalized settings.
>
> **A**: Thank you for this valuable suggestion. We conducted additional experiments in a few-shot in-domain setting on the Cityscapes dataset, using only 1/16 of the labeled training data. As shown in the table below, our method surpasses the Rein baseline, demonstrating its effectiveness not only in domain generalization tasks but also in few-shot in-domain semantic segmentation. In future work, we will further explore the performance of our method under a range of few-shot and fully supervised settings to more comprehensively evaluate its potential in broader in-domain scenarios.
>
> | Method   | Source                  | mIoU     |
> | -------- | ----------------------- | -------- |
> | Rein     | +1/16 of Cityscapes     | 82.5     |
> | **Ours** | **+1/16 of Cityscapes** | **83.6** |

---

### Decision · Program_Chairs · 2025-05-01

**Decision:**

Accept (spotlight poster)

**Comment:**

Hi,

Draft has received overall positive reviews, with 2 weak accept and 2 accept.  Dear authors, please update the draft per recommendations and comments of the reviewers.
Authors have indicated that "We will release all code and weights upon acceptance.", we request that the link to code and model should be part of the camera ready version. Its also requested that the full code to reproduce the results from scratch will be released.

Congratulations.

regards

AC